mathematical modelling

science societies, gender equity, editorial boards, homophily, research awards, unconscious bias

**Author for correspondence:**
Michael J. Plank
e-mail: michael.plank@canterbury.ac.nz

# Gender and societies: a grassroots approach to women in science

Alex James[1,2], Rose Chisnall[1] and Michael J. Plank[1,2]

[1]School of Maths and Stats, University of Canterbury, Christchurch 8140, New Zealand
[2]Te Pūnaha Matatini, a New Zealand Centre of Research Excellence, Christchurch, New Zealand

 AJ, 0000-0002-1543-7139; MJP, 0000-0002-7539-3465

Women are under-represented in science. We show that the extent of the gender gap varies depending on the status of the position in question and there are simple steps that can be taken to improve diversity. We analyse data on the activities of over 30 science societies spanning four countries and five distinct discipline areas. Our results show that women tend to be equally represented in lower status roles and awards, e.g. student prizes and editorships, but under-represented in higher status roles, e.g. late-career awards and chief editorships. We develop a simple mathematical model to explore the role of homophily in decision making and quantify the effect of simple steps that can be taken to improve diversity. We conclude that, when the stakes are low, efforts to tackle historic gender bias towards men have been at least partially successful, but when the stakes are higher male dominance is often still the norm.

## 1. Introduction

Science is sexist [1,2]: the number of women in senior academic positions in many scientific disciplines is disproportionately low [3–5]. This is despite the fact that the proportion of female postgraduate students has been at least 50% in many disciplines for some time [6,7]. The causes of this disparity are numerous and interacting, and it is difficult to separate their relative importance [8]. There are numerous structural factors that can contribute to the underrepresentation of women in senior positions [9]. These include career interruptions for parental leave [10], limited ability to travel due to caregiving responsibilities [2], an unequal division of less prestigious service roles such as mentoring and pastoral care [3,11], and the effects of unconscious bias [1]. Gender imbalance has been shown to have a detrimental effect on productivity, economics and wellbeing [12,13] and many countries have nationwide strategies in place to combat the problem [14]. In this paper, we analyse data on

leadership roles and awards given by scientific societies and propose a simple, mathematical model of an academic award.

Scientific societies play an important role in bringing together scientists in a specific discipline within a particular country. They provide leadership at a national level and are a natural focus point for the discipline. They typically exist externally to and independent from research institutions, such as universities, and can therefore have opportunities to drive change in ways the latter may be unable or unwilling to do. Many societies provide grants for student support and give out awards that are deemed prestigious nationally or internationally. Societies play a wide range of roles but many have a common organizational structure: a leadership body with a president or chief executive who is a research-active scientist; achievement awards that can be categorized as research, teaching or service-based; regular conferences or meetings that encourage student participation; specific student awards often linked to conference presentations; and publication of scientific journals, ranging from highly prestigious international publications to smaller, more local publications.

Many scientific societies publish one or more journals or proceedings. The role of associate editor (referred to by some journals as associate editor, corresponding editor, handling editor or editorial board member), well known in academic circles, is responsible for choosing reviewers, considering reviews and making recommendations to the chief editor. The chief editor typically makes the final decision to reject or publish a given article in the journal. Most academics would recommend that junior colleagues take on the role of associate editor at some point in their career, but they may caution against taking the role too early as it can be time-consuming relative to the opportunities for career advancement it presents. There is evidence that men are over-represented as both associate and chief editors, although some of these studies date back to the 1970s [15] and the disparity has tended to decrease with time [16]. Previous studies have focused on a particular scientific discipline [15,17–19] and used a variety of benchmark statistics as a comparison for women's representation, for example the proportion of women in that discipline area in the United States or United Kingdom [17,19], the female population of 51% [16], and the rate of female representation on similar journals [18]. All these studies conclude that women are under-represented on editorial boards and in particular as chief editors. Society leadership is another area where lack of female representation has been documented. Potvin et al. [20], in a publication entitled 'diversity begets diversity', examined 202 zoological societies worldwide and found that societies with more female board members were more likely to have more women in executive positions and a clear equity and diversity statement. Again, this study did not benchmark the number of women in leadership roles (approximately 30%) against the proportion of women in the discipline.

There are many awards for achievement in science. The Raise project documents scientific awards given to women since 1981. It shows that, of awards available to people of all genders, over 40% have either never been won by a woman or have only rarely been won by a woman. This is combined with the fact that the prizes women receive tend to have less value and prestige [21]. Since 2000, there have been seven female Nobel laureates out of a total of 174 in physics, chemistry, medicine and economics, and only one female Fields Medallist (the highest honour internationally for research in mathematics) out of 18. Most awards do not carry the status of a Nobel Prize or Fields Medal, but are nevertheless important to shaping careers [22] and rewarding excellence. Lincoln et al. [23], in a study of 13 unnamed science society awards, concluded that the number of awards going to women was increasing, but that these were more likely to be service and teaching awards than research awards. However, this study did not attempt to benchmark the data with the number of women eligible for the award, nor take into account any discipline effects, and was confined to the United States. Conversely, a study of the American Geophysics Union [24] concluded that the proportion of late-career awards being awarded to women in that society (11%) was comparable with the proportion of female professors in the discipline (5–9%). Like Lincoln et al. [23], they concluded that female representation was higher for teaching and service awards. Some societies have awards that are for women only, in some cases established partly as a response to underrepresentation of women. However, there is a danger that this contributes to ghettoization [23], where women are primarily represented in roles and awards that are regarded as low prestige or second rate, while the dominance of men in more highly paid positions and higher-prestige awards continues.

There is a large volume of evidence documenting the effects of unconscious bias in evaluations of academic and research quality [1]. For example, scientists have been shown to rank academic CVs more highly, and recommend higher starting salaries, when the CV is randomly assigned a male rather than a female name [25]. Knobloch-Westerwick et al. [26] found that publications by male authors and on topics associated with male researchers were associated with higher scientific quality.

Wold & Wenneras [27] found that peer reviewers rated fellowship applications from men more highly than those from women, relative to objective measures of their quality.

In the studies referred to above, both men and women were found to be biased towards men in their evaluations. Worryingly, Handley *et al.* [28] found that when individuals were asked to rate the quality of research papers, men rated the quality of research into gender bias lower than did women, particularly when the research claimed to find evidence of gender bias. This bias against gender research was stronger in scientists than in the general population. Unconscious bias can also take the form of homophily, i.e. a preference towards other people who share similar characteristics. For example, Freeman & Huang [29] found that researchers are disproportionately likely to co-author publications with other researchers of the same ethnicity. Helmer *et al.* [30] found that male and female journal editors were more likely to appoint reviewers of the same gender as themselves. Booth *et al.* [31] found that job applications from candidates with Anglo-Saxon sounding names received more callbacks than those with diverse names of other origins. Avin *et al.* [32] modelled homophily using a network with preferential attachment and found that this resulted in a glass-ceiling effect for under-represented groups. Homophily can result from cognitive shortcuts, for example: 'cloning'—favourably assessing someone with similar attributes to one's own; or 'snap judgements'—making decisions based on isolated factors such as 'they worked with the same supervisor as me', rather than the evidence as a whole [24]. Conversely, having women in highly visible leadership roles, such as board members [20] or journal editors [18], tends to improve gender equality throughout an organization.

The process by which award winners are selected varies, but there are structures that are broadly common to a range of awards. Typically, nominations for the award are sought and the decision is made by a committee or panel. For some awards, the panel consists partly or exclusively of previous winners of the award in question. In some cases, this may be specified by the formal guidelines relating to the award. More commonly, it may arise as a result of an *ad hoc* decision and the convenience of asking recent award winners to serve on panels. For example, the society for Australia and New Zealand Industrial and Applied Mathematics has an early-career, a mid-career and a late-career award. Each of these awards is decided upon by a panel of three society members, and the latter two typically include at least two previous award winners.

For many awards, nominations may be made by any member of the society or association making the award. In some cases, nominations are invited from previous award winners, or from particular subgroups or institutions. There is evidence that women and other under-represented groups are less likely to be nominated or to self-nominate for awards [24]. The Raise Project specifically advises women to 'Check previous award winners. Is their work similar to yours or that of your nominee?' Although this advice relates to the type of work a potential nominee has undertaken, one can easily imagine it being applied, explicitly or implicitly, to the demographic profile of previous winners. Some societies, for example, the Royal Society (UK) and the New Zealand Mathematical Society, have established nominating committees in an effort to solicit a diverse set of nominees and to ensure under-represented groups are not deterred from being nominated.

In this article, we examine the gender balance of 31 societies, spanning four countries and five disciplines, with a dataset comprising almost 6000 individuals. Our analysis includes leadership and editorial roles, and research awards and prizes given by these societies. We only include awards that are made on the basis of excellence in research and are open to people of all genders. We show that, in many areas, women are represented at a level that is broadly congruent with the proportion of women working in the relevant discipline. However, the roles and awards where representation of women is high are usually the lower status ones; when prestige is at stake, male dominance tends to continue. To complement our empirical analysis, we propose a simple, mathematical model for an academic award. For simplicity, we assume that each individual is characterized by a trait which is described by a single number. Different individuals can have different trait values, representing diversity of members of a particular society or academic community. We use the model to explore the effect on the diversity of award winners of two factors: (i) unconscious bias, in the form of homophily, on the part of panellists or committee members, i.e. they implicitly favour nominees with similar trait values to their own; (ii) bias in the nomination or self-nomination process. We conclude with simple recommendations about the composition of awards panels.

## 2. Material and methods

The range of disciplines studied was chosen to represent a broad spectrum of science, in particular ranging from disciplines with relatively high female representation (ecology) to those with a low

proportion of women (mathematics, economics). The four countries were chosen as all being English-speaking, geographically diverse in position and size, with a strong scientific tradition, good publicly available record keeping and a culture of professional science societies. Web searches were used to find a selection of appropriate societies for each discipline and country.

Society webpages (accessed between December 2017 and February 2018) were used to provide raw data on presidents and award winners since 2000 and society publications. Editorial boards were taken from publishers' webpages. Journal editors were subdivided into two categories: (1) chief editors (including subject editors and editors-in-chief, which are generally higher status roles); (2) associate editors (including handling editors and editorial board members). Summary and raw data were sent to all current society presidents or chief executives for review and correction (June–August 2018). For a summary of the data for each society see electronic supplementary material, table S1.

Gender was inferred manually for each individual in the dataset based on their name; where gender was not immediately obvious (either by a gender-non-specific first name or by only initials being available), web searches were done to find images of the individual concerned. Names that could not be classified this way (less than 0.1%) were excluded. The gender assignment of the raw name data was cross-checked using the web-based name-to-gender inference service, Gender API (http://gender-api.com), which was found to be the best-performing of five such services [33]. Gender API produced the same result as the manual gender classification in 96% of cases, produced a different result in 3% of cases, and no result in 1% of cases.

To compare the proportion of women receiving a given award or role to the prevalence of women in eligible positions, we retrieved benchmark data from a range of sources for the proportion of women in each country-discipline grouping in four categories: (1) senior academics, (2) junior academics, (3) all academics, and (4) postgraduate students (see electronic supplementary material for details). We used (1) as the benchmark for the late-career research award, president and chief editor categories, (2) as the benchmark for the early-career research award category, (3) as the benchmark for the associate editor category, and (4) as the benchmark for the student prize category. These benchmark data do not perfectly represent the number of eligible women for a given award or role. For example, there may be differences between the number of women in a society and the number of women in the discipline; discipline and career-stage boundaries are sometimes arbitrarily defined and may not perfectly correspond to the discipline and career-stage eligibility criteria for a given award or role. Nevertheless, these are the best benchmarking data publicly available.

We calculated risk ratios as the proportion of women receiving the award or role divided by the proportion of women in the relevant benchmarking data for that discipline and country, for each of the six categories of role or award above. Where data were available for more than one society in a particular country-discipline grouping, the data were pooled before the risk ratio was calculated.

# 3. Empirical results

Figure 1 shows the proportion of women, grouped by country and discipline, in six categories of award or role that were common to the majority of societies in our study: president; chief editor; associate editor; late-career research award; early-career research award; best student presentation prize (see electronic supplementary material, table S1 for details). These graphs show raw data for the proportion of women, not normalized against the number of women in the society. Nevertheless, differences among disciplines are apparent from figure 1; mathematics and economics consistently have lower proportions of women than astronomy, ecology and statistics do. The most noticeable trend is that the proportion of women tends to decrease as the status or seniority of the role increases. For example, the average proportion of student prizes going to women in each discipline is between 38% and 91%, and is over 50% in three out of the five disciplines considered (table 1). In contrast, between 13% and 46% of early-career awards and between 6% and 32% of late-career awards go to women.

In cases where there is a gender imbalance, there are at least two possible explanations (which are not mutually exclusive): (i) the imbalance is a consequence of a gender bias in the awards and appointments themselves; or (ii) the imbalance reflects a skew in the underlying proportions of men and women eligible for the award or role. To try and tease apart these two possible explanations, we compared the results in figure 1 to benchmarking data for the proportion of women among postgraduate students and junior and senior academic staff (electronic supplementary material, tables S2–S5) by country and discipline. In line with previous studies, the benchmarking data shows a decreasing

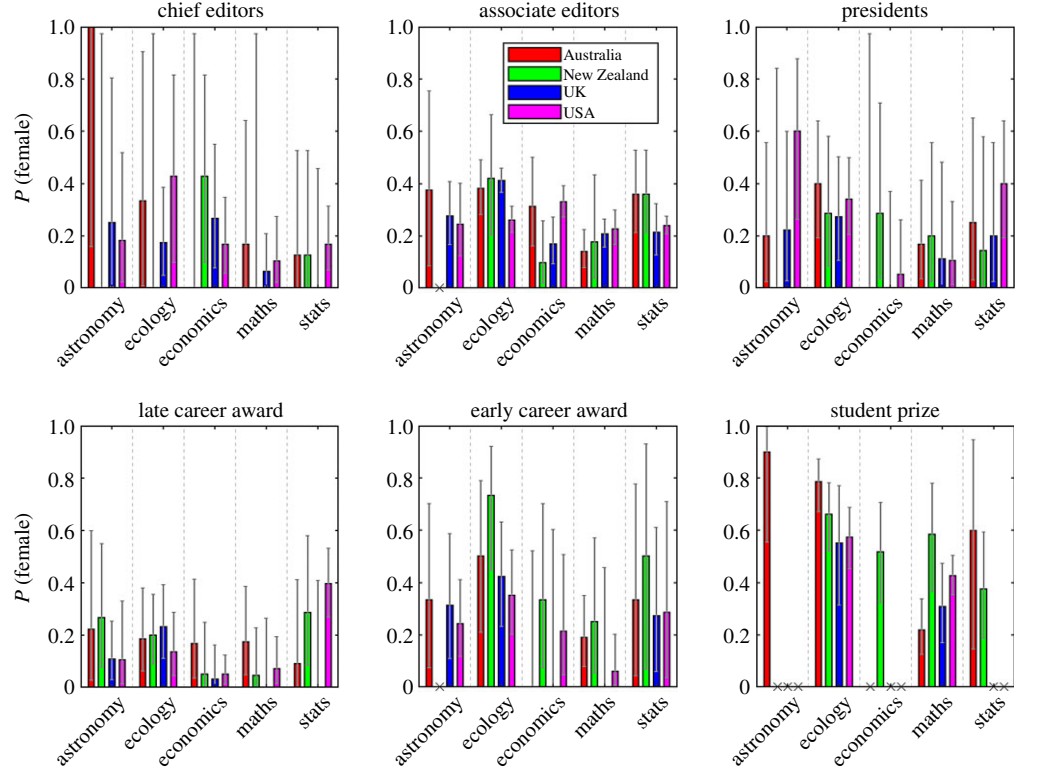

**Figure 1.** High status awards and positions are less likely to be given to women. Proportion of women in each role grouped by country and discipline. Error bars show the binomial distribution 95% confidence interval. Due to small sample sizes, the proportion of women in a given role in a given country and discipline combination is rarely significantly different from 0.5.

**Table 1.** Over a short time frame (less than 30 years), a diverse panel leads to more diversity in winners. Summary results for the model with $P = 3$ panellists and $N = 10$ nominees per year. Diversity $\sigma_n$ is measured as the standard deviation in the award winners' trait values over a period of $n$ years divided by the standard deviation of the population trait values. Results are shown for the short term ($n = 10$ years), medium term ($n = 30$ years) and long term (sufficient time to allow the process to reach a statistical equilibrium). Also shown are the autocorrelation at lag 1 (i.e. correlation between trait values of successive winners) and the probability $P_{50}$ that the winner will come from the central 50% of the trait distribution.

| | | number of previous award winners on the panel $p_w$ | | | |
| --- | --- | --- | --- | --- | --- |
| | | 0 | 1 | 2 | 3 |
| random nominations | long-term diversity $\sigma_\infty$ | 61% | 55% | 48% | 80% |
| | medium-term diversity $\sigma_{30}$ | 61% | 53% | 44% | 27% |
| | short-term diversity $\sigma_{10}$ | 60% | 51% | 39% | 20% |
| | autocorrelation lag 1 | 0.00 | 0.33 | 0.50 | 0.94 |
| | $P_{50}$ | 84% | 88% | 93% | 66% |
| nomination bias | long-term diversity $\sigma_\infty$ | 49% | 44% | 39% | 34% |
| | medium-term diversity $\sigma_{30}$ | 48% | 44% | 37% | 15% |
| | short-term diversity $\sigma_{10}$ | 47% | 42% | 33% | 11% |
| | autocorrelation lag 1 | 0.03 | 0.28 | 0.44 | 0.91 |
| | $P_{50}$ | 94% | 96% | 99% | 99.5% |

trend in the proportion of women at higher career levels, broadly consistent across countries and disciplines (electronic supplementary material, figure S1).

Figure 2 shows the risk ratios $r$, grouped by country and discipline, for the six categories of award or role: $r = 1$ means that the proportion of women receiving the award or role is the same as the proportion

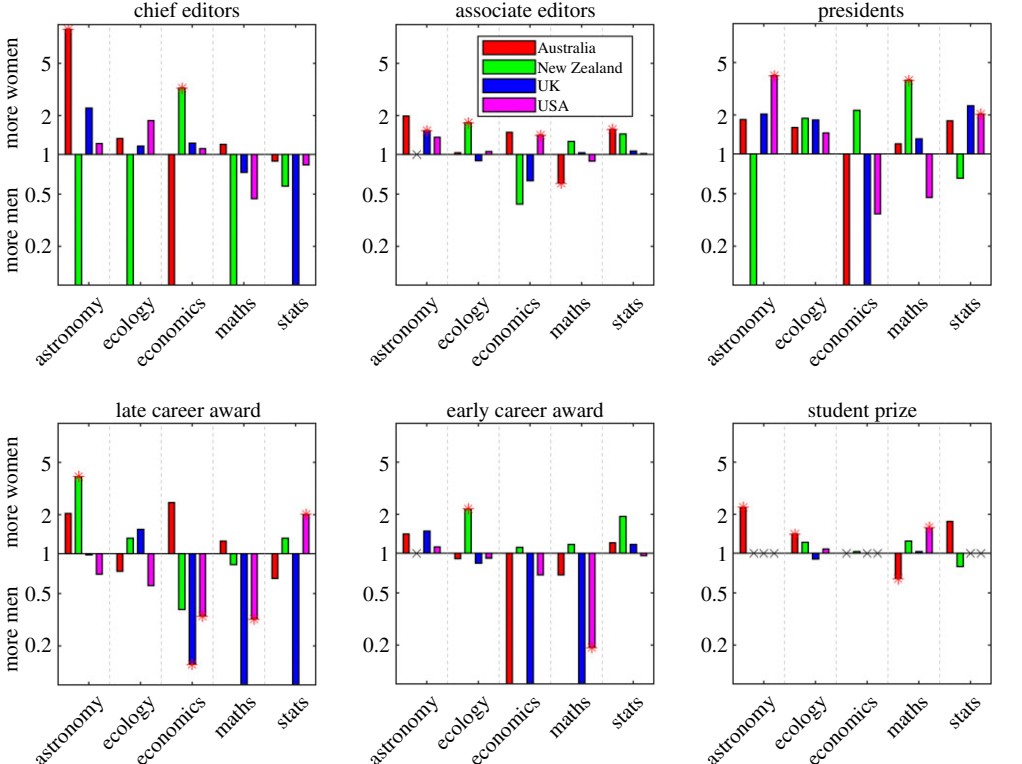

**Figure 2.** Women are, on average, over-represented in lower status awards and positions but under-represented in higher status awards and positions. Risk ratios $r$ (log scale) for each category of award or role and for each country and discipline: $r = 1$ (horizontal line) means that the proportion of women receiving the award or role is the same as the proportion of women at the relevant career stage in that country-discipline grouping; $r < 1$ means that women are under-represented; $r > 1$ means that women are over-represented. Red stars show risk ratios that are significantly different from $r = 1$ (Fisher test $p < 0.05$). Bars that extend to the bottom of the axes indicate cases with no women. X's show country-discipline groupings for which no data were available.

of women at the relevant career stage in that country-discipline grouping; $r < 1$ means that women are under-represented; $r > 1$ means that women are over-represented. Some roles, notably president and associate editor, are less biased towards men than might be expected from previous evidence [15,20]. For most individual country-discipline groupings, the risk ratio is not significantly different from $r = 1$, i.e. there is no significant over/under-representation of women. Pooling the results either by country or by field alone yields no interesting results. However, when the distribution of risk ratios across country-discipline groupings for each category is examined (figure 3), a clearer pattern emerges. The higher the status of the role or award, the more likely it is that women are under-represented. Women are under-represented ($r < 1$) in the associate editor role in only 26% of country-discipline groupings, but are under-represented in the higher status chief editor role in 50% of groupings. These statistics also obscure the fact that, in country-discipline groupings where women are under-represented, the size of the bias tends to be larger than in groupings where women are over-represented, and this is particularly true in more senior awards and roles. For example, only 5% of country-discipline groupings have a risk ratio of $r < 0.5$ (i.e. no or very few women relative to benchmark) in the associate editor role, but 30% of groupings have $r < 0.5$ in the higher status chief editor role. Conversely, no country-discipline groupings have a risk ratio of $r > 2$ (i.e. no or very few men relative to benchmark) in the associate editor role and 15% of groupings have $r > 2$ in the chief editor role. Awards rise in status from student prizes, through early career awards, to late career awards. In student prizes, women tend to dominate and are under-represented in only 25% of country-discipline groupings; in early-career awards, women are under-represented in 53% of groupings; in late-career awards, women are under-represented in 60% of groupings. In addition, for early career awards 20% of groupings have a risk ratio of $r < 0.5$ and this rises to 30% for late-career awards.

Women tend to be over-represented, relative to the benchmark data, in the role of president (risk ratio $r > 1$ in 70% of country-discipline groupings). This role is often considered high status but, in reality,

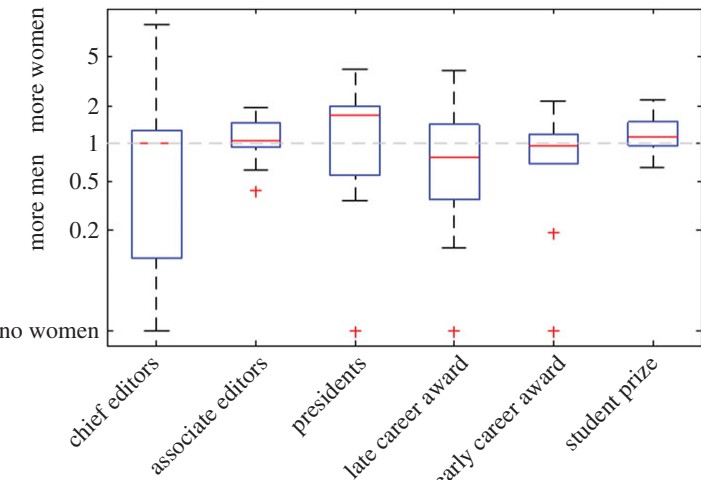

**Figure 3.** Overall women are more likely to be under-represented in higher status roles. Distribution of risk ratios $r$ (log scale) for each category of award or role: $r = 1$ (horizontal dashed line) means that the proportion of women receiving the award or role is the same as the proportion of women at the relevant career stage in that country-discipline grouping; $r < 1$ means that women are under-represented; $r > 1$ means that women are over-represented. On each box, the central mark indicates the median, and the bottom and top edges of the box indicate the 25th and 75th percentiles, respectively. The whiskers extend to the most extreme data points not considered outliers, and the outliers are plotted individually using the '+' symbol.

is also associated with significant administration. However, the over-representation of women in presidential roles should also be considered in light of the small sample sizes in the dataset. For example, 10 of the 31 societies have had either zero or one female president since 2000, and for only three societies is the proportion of female presidents since 2000 higher than 1/3. Since a presidential term is typically two years, for disciplines with very few women at senior levels having a single female president since 2000 is often enough to make them appear over-represented.

## 4. Mathematical model

To provide a theoretical framework for our study, we designed a simple stochastic model of the processes behind many academic research awards. We consider how the composition of the decision-making panel affects the diversity of the award winners. We also test how bias in nominee selection can affect overall diversity. Consider a population of individuals, for example members of particular society, who are eligible for an annual award. We assign all individuals in the population a trait whose value can affect the final decision. This trait is independent of the award criteria and a larger value is not necessarily better than a small value. For example, the trait value could represent a combination of factors associated with age, gender, ethnicity, institutional affiliation, geographical location, and research field. For simplicity, we only consider a one-dimensional trait but this could be readily generalized. We assume that, each year, $N$ nominees make it to the final round of the selection process for the award and there are $P$ panellists that make up the awards panel. We further assume that, at this final stage of the process, all nominees are excellent candidates equally deserving of the award and different panellists could reasonably be expected to favour different nominees. The final decision of the panellists is then subjective: different panels could equally well arrive at different decisions. We assume that all panel members have an unconscious bias towards nominees with a similar trait value to their own, i.e. they would choose the nominee with a trait value closest to their own if possible.

Suppose that, each year, the panel is made up of the $p_w$ most recent previous award winners, and $p_p$ panellists randomly selected from the population of eligible individuals, with $p_p + p_w = P$. For simplicity, we assume that the distribution of trait values across the population is uniform $U[0,1]$. We assume that the award winner for that year is the nominee whose trait value is closest to the mean of the panellists' trait values.

We consider two models for the nomination process. In the first model, the $N$ nominees are selected independently at random from the eligible population. The second model describes a bias in the

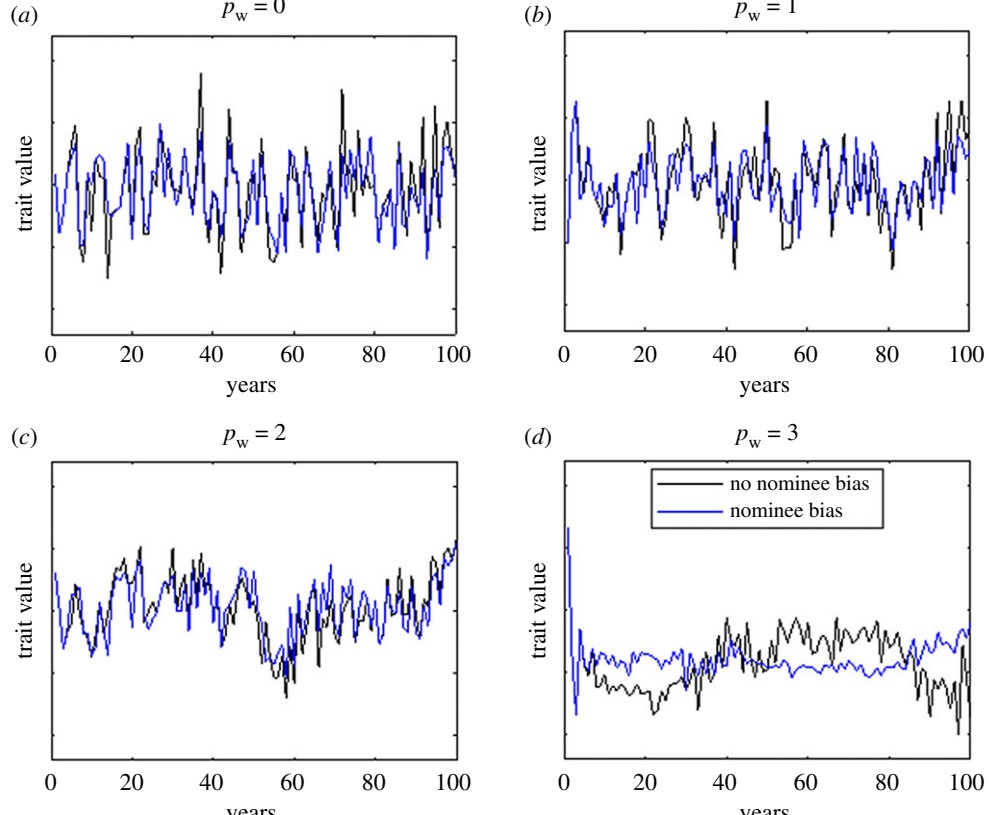

**Figure 4.** The composition of the award panel has a significant effect on the diversity of the winners. (*a*) When panellists are randomly selected from the population each year ($p_w = 0$), there is no autocorrelation in the time series of award winners' trait values. (*d*) When the panel is composed entirely of previous award winners $p_w = P$, there is a strong autocorrelation in the time series, reducing the short-term diversity of award winners. When nomination bias is included in the model (blue lines) this effect is more pronounced. Results are for $P = 3$ panellists and $N = 10$ nominees per year.

nomination process: initially, a pool of $2N$ potential nominees is selected independently at random from the eligible population; then the 50% of potential nominees whose trait values are furthest from the mean of the last 10 award winners are discarded. This represents individuals who may have excellent credentials for the award, but perceive themselves as poor candidates because their trait value is so far removed from those of recent winners.

We simulate time series of the trait values of award winners and examine their properties as the ratio $p_p/p_w$ varies. The first $N$ award winners are randomly selected from the population to initialize the process. Figure 4 shows four example time series of award winners' trait values for a panel of size $P = 3$, with $p_w = 0$, 1, 2, or 3 previous winners, with and without nomination bias. It is clear that, when the panel consists exclusively of previous award winners ($p_w = P$), the time series has very strong autocorrelation (autocorrelation at lag 1 $R_1 = 0.94$) (figure 4*d*), i.e. the trait value of the winner in year $i$ is very similar to that of the winner in year $i + 1$. This means that, in the short term, there is a lack of diversity in the award winners. At the other end of the spectrum, when all panellists are randomly selected from the general population ($p_w = 0$) and there is no nomination bias, the winners' trait values are not correlated from year to year (figure 4*a*, black line, $R_1 = 0.01$). Nomination bias has little impact on the results when the panellists are randomly selected from the general population ($p_w = 0$, figure 4*a*, blue line) because bias in the nominees is mitigated by diversity in recent award winners. Conversely, when the panel consists exclusively of previous winners ($p_w = P$), nomination bias further reduces the diversity of award winners (figure 4*d*, blue line).

To quantify the diversity in the award winners, we calculated the mean and standard deviation of the award winners' trait values. Over a long time period, the mean of the winners' trait values is the same as the mean of the population, which in this case is 0.5. We measure diversity $\sigma$ as the standard deviation of the winners' trait values relative to the standard deviation of the population trait values (table 1). When all panellists are randomly selected ($p_w = 0$), the award winners' trait values have a long-term diversity

of 60%. This means that there is a substantial proportion of the population that is unlikely to ever win an award because their trait value is too far away from that of an average panellist. Conversely, in the very long term, a panel made up entirely of previous award winners ($p_w = P$) gives the most diverse selection of award winners. This is because the strong autocorrelation in winners' trait values dominates the tendency to revert to the population mean, allowing the mean panel trait value eventually to sample almost all of the population distribution.

However, the long-term results described above do not capture the whole story: it can take hundreds of years for the stochastic process to reach statistical equilibrium. Hence, we calculated the standard deviation of winners' trait values over a period of $n$ years, relative to the population standard deviation, averaged over multiple realizations of the process. In the short term ($n = 10$ years), the panel of previous award winners ($p_w = P$) results in very low diversity of winners relative to the diversity of the population: 21% when there is no nomination bias, and only 10% when there is nomination bias (table 1). In the medium term ($n = 30$ years), this diversity increases slightly to 27% with no nomination bias and 14% with nomination bias. In contrast, when all panellists are randomly selected ($p_w = 0$), the short-term diversity is the same as the long-term diversity (approximately 60% without nomination bias and 47% with nomination bias). Mixed panels ($0 < p_w < P$) result in intermediate levels of short- and medium-term diversity.

Increasing the panel size $P$ decreases the diversity of award winners in the model. As the panel size increases, the mean panel trait value becomes more tightly distributed around the population mean. This makes it harder and harder for nominees with trait values at the margins of the distribution to win the award. As a result, award winners become more and more homogeneous. This phenomenon occurs almost universally regardless of the composition of the panel.

Increasing the number of nominees $N$ has a less uniform effect. When all panellists are randomly selected ($p_w = 0$) and there is no nomination bias, increasing $N$ results in a small decrease in diversity. For example, increasing $N$ from 10 to 50 results in a decrease of three percentage points in all three diversity measures. When $p_w = 0$ and there is nomination bias, increasing $N$ slightly increases diversity. Conversely, for the panel of previous award winners ($p_w = P$), a larger nominee pool results in significantly less diversity, e.g. increasing $N$ from 10 to 50 reduces short-term diversity with and without nomination bias to just 2% and 4%, respectively.

# 5. Discussion

Partly in response to increasing data and research on the gender gap in science, many science societies have taken initiatives to increase the representation and recognition of under-represented groups, including women. For example, an increasing number of societies (e.g. the Royal Society (UK), the New Zealand Mathematical Society) have nomination committees, which are responsible for soliciting a diverse and representative range of nominees for Society awards, fellowships and leadership positions. Many society conferences have special interest subgroups, meetings or sessions dedicated to women in science (e.g. American Economic Association, Australian and NZ Industrial and Applied Mathematics). Financial support, such as travel grants aimed at women and/or carers (e.g. American, Australian and NZ Mathematical Societies) aims to remove barriers to female participation at conferences. Societies are increasingly adopting codes of conduct, requiring all participants at a conference to agree to a standard of behaviour and providing avenues for redress in situations where the code of conduct is contravened (e.g. Astronomical Society of Australia, NZ Mathematical Society). Some societies now have sponsorship policies that explicitly make financial support conditional on satisfactory representation of women among invited speakers and conference organizing committee members, or on adopting other measures outlined here such as a code of conduct (e.g. Australia and NZ Industrial and Applied Mathematics, Astronomical Society of Australia).

Overall, our results suggest that these initiatives have been partially successful in redressing historical gender biases. However, this success is predominantly confined to early career stages and lower prestige roles and awards; there remains much work to be done to achieve gender equity in the higher echelons of science academia. Scientific societies, in their capacity independent from universities and other employers, are well placed to take bottom-up, grassroots approaches to achieving this aim.

Our model results show that having an awards panel wholly or largely composed of previous award winners can reduce the diversity of award winners in the short and medium term. This occurs because the panellists tend to be restricted to a small subset of the population trait space for prolonged periods of time, meaning that only those population members whose characteristics are currently in vogue are

likely to win the award. Our results also show that nomination bias can be a factor in reducing diversity of award winners. Some of the organizations surveyed have made efforts to counter this by establishing nominating committees. This can be an effective way of ensuring well-qualified applicants from under-represented groups are not deterred from being nominated and soliciting a broad pool of nominees.

Our simplified model only considers bias arising from homophily, i.e. the preference of panellists for nominees who share similar characteristics to themselves. The model predicts that, in the long term, a panel consisting of previous award winners, will eventually drift around the trait space sufficiently to sample the majority of the population diversity (although this may take hundreds of years). However, in reality, unconscious or implicit bias of the panellists is combined with a range of systemic barriers to women and other historically under-represented groups. For example, women are under-represented in academic leaderships and decision-making roles, such as editorial boards [16,17,19], peer reviewers [30] and professorships [4,5]. Women publish fewer papers than men, receive less research funding, are less likely to be first authors, have fewer international collaborations, and are more likely to have career interruptions [2]. Such external structural biases are likely to limit 'panel drift' and to restrict the panellists and award winners to the same unrepresentative subset of the population *ad infinitum*. These factors are also likely to contribute to the imbalances in senior leadership positions. For example, a long, unbroken period of service as an Associate Editor may lead to promotion to Chief Editor, although it is not possible to test this with the current dataset because it only contains editorial roles at a single time point.

In many cases, it is desirable for committees to retain some institutional memory and detailed knowledge of award criteria from one year to the next. This may make it undesirable to have a completely new panel selected from the general population each year. A sensible compromise is to have a panel consisting of a mixture of previous award winners and general committee members. Our results show that having a committee of one previous award winner and two randomly selected members gives almost as high a diversity of award winners as a committee of three randomly selected members. Another practical step would be to have the committee members serve on a rolling basis for a number of years, so that only one committee member has to be replaced per year.

For simplicity, we assumed that general committee members were selected randomly from the population, and that the award winner was the nominee whose trait value was closest to the mean of the panellists'. An obvious attempt to improve diversity of the panel would be to actively seek a selection of panellists with a range of different trait values. In our simplified model, this would not necessarily improve the diversity of award winners because the mean of a diverse selection of panellists is still likely to be close to the population mean, especially with a large panel. In this sense, smaller panels may be better for diversity than larger ones, because they will have more variation in their mean from one year to another. However, it is a limitation of the simple model that the nominee with the trait value closest to the panel mean is always selected as the winner.

One drawback of the modelling approach we have taken is the representation of a wide range of individual traits by a single number. At first this simplification seems unrealistic but we argue that although individuals' traits are multi-faceted they are also highly correlated. For example, subject area affects not only the type and quantity of research outputs but also the sources of funding and size of funding applications, e.g. anecdotally pure mathematicians publish less frequently than applied mathematicians, who in turn are more likely to have joint author publications and possibly larger funding awards when technical equipment is needed. Of course, it would be possible to generalize the model to include multiple traits. However, it is likely this would produce similar predictions (having previous winners on the panel tends to reduce short- to medium-term diversity in award winners) without any real additional insight.

# 6. Recommendations

We conclude with two simple recommendations for award-granting organizations:

1. Include a diverse range of people on the selection panel and avoid having panels consisting mainly or exclusively of previous award winners.
2. Take steps to solicit a diverse a range of nominees for awards, such as establishing a nominations committee.
3. Publish and maintain a list of selection panel members.

The first two recommendations are informed by the results of our model and are broadly consistent with previous recommendations for redressing gender imbalance in academic awards [34]. The third recommendation would provide useful data for testing and refining the model presented here, as well as allowing societies to monitor the gender balance of their selection panels over time.

Data accessibility. Data are available in electronic supplementary material, table S1.
Authors' contributions. R.C. acquired the raw data. A.J., R.C. and M.J.P. performed the conceptual design and data analysis. A.J. and M.J.P. developed the methodology and wrote the first and subsequent drafts of the manuscript.
Competing interests. The authors declare no competing interests.
Funding. Financial support came from Te Pūnaha Matatini, a New Zealand Centre of Research Excellence.
Acknowledgements. We are grateful to the presidents and chief executives who gave feedback on the summary data for their society, to Te Pūnaha Matatini for financial support, and to anonymous reviewers for comments on earlier versions of this manuscript.

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
