## [Reviewer comments · Royal Society Open Science]

Review History

RSOS-190633.R0 (Original submission)

Review form: Reviewer 1

Is the manuscript scientifically sound in its present form?

Yes

Are the interpretations and conclusions justified by the results?

Yes

Is the language acceptable?

Yes

Is it clear how to access all supporting data?

No

Do you have any ethical concerns with this paper?

No

Have you any concerns about statistical analyses in this paper?

No

Recommendation?

Accept with minor revision (please list in comments)

Comments to the Author(s)

This paper considers the status of women in scientific societies in New Zealand, Australia, the UK and the USA in two ways. Firstly, the authors have collected publicly available data on the proportion of women who hold prestigious positions within these societies, including president, editorial roles, and medal winners. They use this data to show that, for instance, the more prestigious a position is, the less likely it is that this role reflects the gender balance within the society itself i.e. men are still advantaged when it comes to more prestigious positions. Secondly, they construct a model that demonstrates how homophily can generate these kind of outcomes in medalist selection panels if previous winners are allocated to the panel. Both strands of work are very interesting in their own right, but together they allow the authors to make some useful recommendations for scientific societies that want to improve diversity in their senior ranks. The paper is very well written, easy to follow, and the methodology appears robust. It is highly topical and will no doubt find a wide audience.

I recommend publication, after the authors consider the following comments:

* The authors have focused on bias and homophily as causes for underrepresentation, but they should at least discuss other possible structural causes. For instance, I wonder about the influence of career breaks, particularly when it comes to parental leave, which are still more likely to affect the careers of women than men (e.g. Ecklund and Lincoln 2011). A lengthy, unbroken stint as Associate Editor may be more likely to result in promotion to Editor, for instance. Is it possible to look at this with their dataset?

* Likewise, although this may be harder to test with the data they have, I wonder about the influence of the second shift in STEM, whereby men avoid, and women take on, a greater share of the less prestigious service roles, such as pastoral care of students (Misra, Joya, et al. 2011). This may lead to accumulated advantage that may translate into awards and medals, compounding bias in selection panels that are unaware or dismissive of lower prestige service roles.

* I wonder if a further recommendation might be worthwhile: that societies publish and maintain a list of the names of selection panels. I am aware that my society puts the names of panel members on its website every year, but I don't believe it maintains a longitudinal record of these panels. This kind of data would be invaluable for testing and improving on the model presented here, and allow societies to monitor

Minor comments

* I struggled to understand line 87-88 on pg 3: "40% of awards open to people of all genders ... have only rarely been given to women ..."

* Royal Society of New Zealand is now Royal Society Te Apārangi. Similarly I wonder if the Royal Society on line 153 means Royal Society Te Apārangi.

Review form: Reviewer 2**Is the manuscript scientifically sound in its present form?**

Yes

Are the interpretations and conclusions justified by the results?

Yes

Is the language acceptable?

Yes

Is it clear how to access all supporting data?

Yes

Do you have any ethical concerns with this paper?

No

Have you any concerns about statistical analyses in this paper?

No

Recommendation?

Accept with minor revision (please list in comments)

Comments to the Author(s)

This paper presents a statistical model that shows strong evidence of homophily in science, that is, the tendency to form strong social connections with and reward people who share one's defining characteristics. The model is useful because it predicts and compares outcomes from award committees that are comprised exclusively of past award winners with those that are less so.

To the best of my knowledge, this is the first time such a study has been conducted based on data from professional societies. It covers five discipline areas and four English-speaking countries. It is a very strong study and, in general, it is well written. I therefore recommend it for publication, but point out possible improvements in the exposition and minor errors that should be corrected before it is published.

+ page 2, line 91: Wikipedia points out that 60 people have been awarded Fields medals so far, not 18.

+ page 5, line 156: I would like to see the statement made here: "we propose a simple, mathematical model for an academic award" earlier in the paper. Previous paragraphs focused mostly on a literature review. It would be valuable to state what is new here earlier in the paper.

+ page 5, line 158: I suggest replacing the phrase "Everybody's trait value is different to represent" by "Everybody's trait value is allowed to vary in order to represent".

+ page 6, line 179: It sounds a little awkward to say "Each individual in the dataset was assigned a gender manually based on their name". Perhaps this should be "Gender was inferred for each individual in the dataset from their name".

+ page 15, Figure 1: It took me a moment to realise that the data presented in the top row of figures were grouped by discipline in the same way as the bottom row. To increase clarity, it would be good practice to also display the discipline labels for the top row. The same comment applies to Figure 2 on page 16.

+ page 17, Figure 3: Could the abbreviations "C-Ed", "Ed", "Pres", "LCA", "ECA", "Stud" be explained in the caption? This would be helpful for readers from non-English speaking backgrounds.

+ pages 19-22: Figures 1-4 are repeated without figure labels in the captions. Was this intentional?

Review form: Reviewer 3

Is the manuscript scientifically sound in its present form?

Yes

Are the interpretations and conclusions justified by the results?

No

Is the language acceptable?

No

Is it clear how to access all supporting data?

Yes

Do you have any ethical concerns with this paper?

No

Have you any concerns about statistical analyses in this paper?

I do not feel qualified to assess the statistics

Recommendation?

Accept with minor revision (please list in comments)

Comments to the Author(s)

I was undecided whether to list this review as Minor or Major revisions. I agree that it's useful to analyse the difference between male and female roles in scientific societies, and the authors have clearly gathered a lot of evidence, and completed the data with some modelling.

Overall, the manuscript was well-written, but precision was an issue in places, as discussed below. Presentation of the data, and conclusions drawn from the data presented, also need improvement.

I had the following concerns, which I believe need to be addressed before publication in this journal:

Point 1: Meaning of the data: I have some issues with the way the data was presented, and some of the conclusions drawn from the data. E.g.

1. Figure 1: This data is not normalized against the number of women in the societies, and hence doesn't show much useful information, I suggest remove. The data in Figures 2-3 is much more useful, since normalized against the numbers of women, but there are still serious issues, as outlined below.

2. Why no error bars or other symbols to demonstrate variability on Figure 2? Also bar graph seems a strange format for this kind of data?

3. Figure 2 presents results by field and by country, Figure 3 pools them completely. What about when pooled just by field across all countries? Or by country across all fields? If pooling in this way doesn't show much, then even that is useful, it could be reported in the text. For example, (Ceci, Ginther et al. 2014) found that women actually do better in male-dominated fields of science, how does your data compare to their findings? This would be a useful comparison, because you have both male-dominated and non-male-dominated disciplines of science

4. One of your key conclusions is that women are under-represented in high status roles, I'm not convinced of this based on your data and discussion. You define chief editor as higher status than editor, but what about president? Furthermore, the differences data for chief and editors in figure 2 also don't easily lead me to conclude that men are more likely to be chief editor. See comment above also, about what this data would look like if pooled across discipline and /or nation. You also need some discussion of what is the status of editor vs president. Is president low status? Classify your positions each specifically as high or low status, consider even showing that classification on the plots

5. Figure 3: Your title includes the statement "...though this is improving", but that statement is not supported by the data, because you haven't presented data over time, or referred to other time/data

6. You could easier make the x axis on this plot easier to read, using full names

Point 2: Imprecision

1. Introduction was in general well-written, but the first paragraph rather sensational, and imprecise in places, eg. "Science is sexist". I've read lots of papers about gender in science, but I can't agree with this statement, particularly because you haven't defined what you mean by sexist. If you want to make this statement, back it up with reasoning, including a definition of sexist

2. Throughout the introduction, you switch numerous times back and forth between the literature, and this study. To see this, do a search on "our analysis", and you will see that in some paragraphs you switch back and forth a number of times between what is known, and what you are trying to do here, which makes it hard for the reader to get a really coherent picture of the purpose of this paper, and its contribution

Other issues of precision in writing:

L65: Is what you call "Associate Editor" here what you call "Editor" in the figures? Best to keep terminology clear and consistent.

L95-97: reverse order of the sentence to make it clearer: women are getting more awards, but more likely to get those awards for teaching and service.

L149: reference needed

L190: I presume you normalize by the number of women in the society, pooled by societies in that discipline across the country? Note that numbers in the society may not equal numbers in the discipline.

L221-222: reference needed

L398: Second recommendation is written as "either or", but these are 2 quite different statements with different consequences

Ceci, S. J., D. K. Ginther, S. Kahn and W. M. Williams (2014). "Women in academic science: a changing landscape." *Psychological Science in the Public Interest* 15(3): 75-141.

1.

Decision letter (RSOS-190633.R0)

10-Jun-2019

Dear Dr Plank

On behalf of the Editors, I am pleased to inform you that your Manuscript RSOS-190633 entitled "Gender and societies: a grassroots approach to women in science" has been accepted for publication in Royal Society Open Science subject to minor revision in accordance with the referee suggestions. Please find the referees' comments at the end of this email.

The reviewers and handling editors have recommended publication, but also suggest some minor revisions to your manuscript. Therefore, I invite you to respond to the comments and revise your manuscript.

- Ethics statement

- Data accessibility

If you wish to submit your supporting data or code to Dryad (<http://datadryad.org/>), or modify your current submission to dryad, please use the following link:
<http://datadryad.org/submit?journalID=RSOS&manu=RSOS-190633>

- Competing interests

- Authors' contributions

- Acknowledgements

- Funding statement

Because the schedule for publication is very tight, it is a condition of publication that you submit the revised version of your manuscript before 19-Jun-2019. Please note that the revision deadline will expire at 00.00am on this date. If you do not think you will be able to meet this date please let me know immediately.

on behalf of Mark Chaplain (Subject Editor)
openscience@royalsociety.org

Associate Editor Comments to Author:

Three reviewers have commented on your paper, each broadly recommending publication after you've performed a number of modifications (most of which appear to be relatively minor). Please ensure you fully address these matters in your revision, and include a point-by-point response to the queries raised.

Reviewer comments to Author:
Reviewer: 1

Comments to the Author(s)

This paper considers the status of women in scientific societies in New Zealand, Australia, the UK and the USA in two ways. Firstly, the authors have collected publicly available data on the proportion of women who hold prestigious positions within these societies, including president, editorial roles, and medal winners. They use this data to show that, for instance, the more prestigious a position is, the less likely it is that this role reflects the gender balance within the society itself i.e. men are still advantaged when it comes to more prestigious positions. Secondly, they construct a model that demonstrates how homophily can generate these kind of outcomes in medalist selection panels if previous winners are allocated to the panel. Both strands of work are very interesting in their own right, but together they allow the authors to make some useful recommendations for scientific societies that want to improve diversity in their senior ranks. The paper is very well written, easy to follow, and the methodology appears robust. It is highly topical and will no doubt find a wide audience.

I recommend publication, after the authors consider the following comments:

* The authors have focused on bias and homophily as causes for underrepresentation, but they should at least discuss other possible structural causes. For instance, I wonder about the influence of career breaks, particularly when it comes to parental leave, which are still more likely to affect the careers of women than men (e.g. Ecklund and Lincoln 2011). A lengthy, unbroken stint as Associate Editor may be more likely to result in promotion to Editor, for instance. Is it possible to look at this with their dataset?

* Likewise, although this may be harder to test with the data they have, I wonder about the influence of the second shift in STEM, whereby men avoid, and women take on, a greater share of the less prestigious service roles, such as pastoral care of students (Misra, Joya, et al. 2011). This may lead to accumulated advantage that may translate into awards and medals, compounding bias in selection panels that are unaware or dismissive of lower prestige service roles.

* I wonder if a further recommendation might be worthwhile: that societies publish and maintain a list of the names of selection panels. I am aware that my society puts the names of panel members on its website every year, but I don't believe it maintains a longitudinal record of these panels. This kind of data would be invaluable for testing and improving on the model presented here, and allow societies to monitor

Minor comments

* I struggled to understand line 87-88 on pg 3: "40% of awards open to people of all genders ... have only rarely been given to women ..."

* Royal Society of New Zealand is now Royal Society Te Apārangi. Similarly I wonder if the Royal Society on line 153 means Royal Society Te Apārangi.

Reviewer: 2

Comments to the Author(s)

This paper presents a statistical model that shows strong evidence of homophily in science, that is, the tendency to form strong social connections with and reward people who share one's defining characteristics. The model is useful because it predicts and compares outcomes from award committees that are comprised exclusively of past award winners with those that are less so.

To the best of my knowledge, this is the first time such a study has been conducted based on data from professional societies. It covers five discipline areas and four English-speaking countries. It is a very strong study and, in general, it is well written. I therefore recommend it for publication, but point out possible improvements in the exposition and minor errors that should be corrected before it is published.

+ page 2, line 91: Wikipedia points out that 60 people have been awarded Fields medals so far, not 18.

+ page 5, line 156: I would like to see the statement made here: "we propose a simple, mathematical model for an academic award" earlier in the paper. Previous paragraphs focused mostly on a literature review. It would be valuable to state what is new here earlier in the paper.

+ page 5, line 158: I suggest replacing the phrase "Everybody's trait value is different to represent" by "Everybody's trait value is allowed to vary in order to represent".

+ page 6, line 179: It sounds a little awkward to say "Each individual in the dataset was assigned a gender manually based on their name". Perhaps this should be "Gender was inferred for each individual in the dataset from their name".

+ page 15, Figure 1: It took me a moment to realise that the data presented in the top row of figures were grouped by discipline in the same way as the bottom row. To increase clarity, it would be good practice to also display the discipline labels for the top row. The same comment applies to Figure 2 on page 16.

+ page 17, Figure 3: Could the abbreviations "C-Ed", "Ed", "Pres", "LCA", "ECA", "Stud" be explained in the caption? This would be helpful for readers from non-English speaking backgrounds.

+ pages 19-22: Figures 1-4 are repeated without figure labels in the captions. Was this intentional?

Reviewer: 3

Comments to the Author(s)

I was undecided whether to list this review as Minor or Major revisions. I agree that it's useful to analyse the difference between male and female roles in scientific societies, and the authors have clearly gathered a lot of evidence, and completed the data with some modelling.

Overall, the manuscript was well-written, but precision was an issue in places, as discussed below. Presentation of the data, and conclusions drawn from the data presented, also need improvement.

I had the following concerns, which I believe need to be addressed before publication in this journal:

Point 1: Meaning of the data: I have some issues with the way the data was presented, and some of the conclusions drawn from the data. E.g.

1. Figure 1: This data is not normalized against the number of women in the societies, and hence doesn't show much useful information, I suggest remove. The data in Figures 2-3 is much more useful, since normalized against the numbers of women, but there are still serious issues, as outlined below.

2. Why no error bars or other symbols to demonstrate variability on Figure 2? Also bar graph seems a strange format for this kind of data?

3. Figure 2 presents results by field and by country, Figure 3 pools them completely. What about when pooled just by field across all countries? Or by country across all fields? If pooling in this way doesn't show much, then even that is useful, it could be reported in the text. For example, (Ceci, Ginther et al. 2014) found that women actually do better in male-dominated fields of science, how does your data compare to their findings? This would be a useful comparison, because you have both male-dominated and non-male-dominated disciplines of science

4. One of your key conclusions is that women are under-represented in high status roles, I'm not convinced of this based on your data and discussion. You define chief editor as higher status than editor, but what about president? Furthermore, the differences data for chief and editors in figure 2 also don't easily lead me to conclude that men are more likely to be chief editor. See comment above also, about what this data would look like if pooled across discipline and /or nation. You also need some discussion of what is the status of editor vs president. Is president low status? Classify your positions each specifically as high or low status, consider even showing that classification on the plots

5. Figure 3: Your title includes the statement "...though this is improving", but that statement is not supported by the data, because you haven't presented data over time, or referred to other time/data

6. You could easier make the x axis on this plot easier to read, using full names

Point 2: Imprecision

1. Introduction was in general well-written, but the first paragraph rather sensational, and imprecise in places, eg. "Science is sexist". I've read lots of papers about gender in science, but I can't agree with this statement, particularly because you haven't defined what you mean by

sexist. If you want to make this statement, back it up with reasoning, including a definition of sexist

2. Throughout the introduction, you switch numerous times back and forth between the literature, and this study. To see this, do a search on “our analysis”, and you will see that in some paragraphs you switch back and forth a number of times between what is known, and what you are trying to do here, which makes it hard for the reader to get a really coherent picture of the purpose of this paper, and its contribution

Other issues of precision in writing:

L65: Is what you call “Associate Editor” here what you call “Editor” in the figures? Best to keep terminology clear and consistent.

L95-97: reverse order of the sentence to make it clearer: women are getting more awards, but more likely to get those awards for teaching and service.

L149: reference needed

L190: I presume you normalize by the number of women in the society, pooled by societies in that discipline across the country? Note that numbers in the society may not equal numbers in the discipline.

L221-222: reference needed

L398: Second recommendation is written as “either or”, but these are 2 quite different statements with different consequences

Ceci, S. J., D. K. Ginther, S. Kahn and W. M. Williams (2014). "Women in academic science: a changing landscape." *Psychological Science in the Public Interest* 15(3): 75-141.

1.

Author's Response to Decision Letter for (RSOS-190633.R0)

See Appendix A.

RSOS-190633.R1 (Revision)

Review form: Reviewer 1

Is the manuscript scientifically sound in its present form?

Yes

Are the interpretations and conclusions justified by the results?

Yes

Is the language acceptable?

Yes

Do you have any ethical concerns with this paper?

No

Have you any concerns about statistical analyses in this paper?

No

Recommendation?

Accept as is

Comments to the Author(s)

The authors have addressed my concerns and suggestions in their revision. I recommend that it be accepted as is.

Review form: Reviewer 3

Is the manuscript scientifically sound in its present form?

Yes

Are the interpretations and conclusions justified by the results?

Yes

Is the language acceptable?

Yes

Do you have any ethical concerns with this paper?

No

Have you any concerns about statistical analyses in this paper?

No

Recommendation?

Accept with minor revision (please list in comments)

Comments to the Author(s)

Great response to reviewers, you have addressed all concerns. Paper reads very well. Minor (optional) point: rotate the x labels on Figure 3, as you have done in Figures 1-2, and write the titles in full. You can easily fit them, will make it much more readable. Up to the editor if they think this is worthwhile.

Decision letter (RSOS-190633.R1)

26-Jul-2019

Dear Dr Plank,

I am pleased to inform you that your manuscript entitled "Gender and societies: a grassroots approach to women in science" is now accepted for publication in Royal Society Open Science.

You can expect to receive a proof of your article in the near future. Please contact the editorial office (openscience_proofs@royalsociety.org and openscience@royalsociety.org) to let us know if

you are likely to be away from e-mail contact. Due to rapid publication and an extremely tight schedule, if comments are not received, your paper may experience a delay in publication.

Please note the Editors would like you to adhere to the suggestions made by one of the reviewers to rotate the x labels on Figure 3, as done in Figures 1-2, and write the titles in full. Please ensure you include these tweaks during the proofing process.

on behalf of Prof Mark Chaplain (Subject Editor)
openscience@royalsociety.org

Reviewer comments to Author:
Reviewer: 1

Comments to the Author(s)
The authors have addressed my concerns and suggestions in their revision. I recommend that it be accepted as is.

Reviewer: 3

Comments to the Author(s)
Great response to reviewers, you have addressed all concerns. Paper reads very well. Minor (optional) point: rotate the x labels on Figure 3, as you have done in Figures 1-2, and write the titles in full. You can easily fit them, will make it much more readable. Up to the editor if they think this is worthwhile.

Appendix A

Reviewer: 1

Comments to the Author(s)

This paper considers the status of women in scientific societies in New Zealand, Australia, the UK and the USA in two ways. Firstly, the authors have collected publicly available data on the proportion of women who hold prestigious positions within these societies, including president, editorial roles, and medal winners. They use this data to show that, for instance, the more prestigious a position is, the less likely it is that this role reflects the gender balance within the society itself i.e. men are still advantaged when it comes to more prestigious positions. Secondly, they construct a model that demonstrates how homophily can generate these kind of outcomes in medalist selection panels if previous winners are allocated to the panel. Both strands of work are very interesting in their own right, but together they allow the authors to make some useful recommendations for scientific societies that want to improve diversity in their senior ranks. The paper is very well written, easy to follow, and the methodology appears robust. It is highly topical and will no doubt find a wide audience.

I recommend publication, after the authors consider the following comments:

* The authors have focused on bias and homophily as causes for underrepresentation, but they should at least discuss other possible structural causes. For instance, I wonder about the influence of career breaks, particularly when it comes to parental leave, which are still more likely to affect the careers of women than men (e.g. Ecklund and Lincoln 2011). A lengthy, unbroken stint as Associate Editor may be more likely to result in promotion to Editor, for instance. Is it possible to look at this with their dataset?

Thank you for highlighting these issues and references. We have added some text about these points in in the Introduction (Lines 43-47). We have also added a comment in the Discussion about the route from Associate to Chief Editor (lines 386-389). Unfortunately, it is not possible to test this with the dataset because we only have data on editors at a single time point.

* Likewise, although this may be harder to test with the data they have, I wonder about the influence of the second shift in STEM, whereby men avoid, and women take on, a greater share of the less prestigious service roles, such as pastoral care of students (Misra, Joya, et al. 2011). This may lead to accumulated advantage that may translate into awards and medals, compounding bias in selection panels that are unaware or dismissive of lower prestige service roles.

We have also added some text to the introduction about this point (Lines 47-48).

* I wonder if a further recommendation might be worthwhile: that societies publish and maintain a list of the names of selection panels. I am aware that my society puts the names of panel members on its website every year, but I don't believe it maintains a longitudinal record of these panels. This kind of data would be invaluable for testing and improving on the model presented here, and allow societies to monitor

This is an excellent suggestion and we have added this recommendation along with some brief commentary to the conclusion of our manuscript (Recommendation #3 and lines 427-428).

Minor comments

* I struggled to understand line 87-88 on pg 3: "40% of awards open to people of all genders ... have only rarely been given to women ..."

This sentence has been reworded for clarity.

* Royal Society of New Zealand is now Royal Society Te Apārangi. Similarly I wonder if the Royal Society on line 153 means Royal Society Te Apārangi.

This sentence refers to the Royal Society (UK), not the Royal Society Te Apārangi and we have now clarified this.

Reviewer: 2

Comments to the Author(s)

This paper presents a statistical model that shows strong evidence of homophily in science, that is, the tendency to form strong social connections with and reward people who share one's defining characteristics. The model is useful because it predicts and compares outcomes from award committees that are comprised exclusively of past award winners with those that are less so.

To the best of my knowledge, this is the first time such a study has been conducted based on data from professional societies. It covers five discipline areas and four English-speaking countries. It is a very strong study and, in general, it is well written. I therefore recommend it for publication, but point out possible improvements in the exposition and minor errors that should be corrected before it is published.

+ page 2, line 91: Wikipedia points out that 60 people have been awarded Fields medals so far, not 18. **These statistics are for the winners of these medals since 2000. This has been clarified in the text (lines 91-92).**

+ page 5, line 156: I would like to see the statement made here: "we propose a simple, mathematical model for an academic award" earlier in the paper. Previous paragraphs focused mostly on a literature review. It would be valuable to state what is new here earlier in the paper.

We have added a sentence to the end of the opening paragraph to state what is new in the paper.

+ page 5, line 158: I suggest replacing the phrase "Everybody's trait value is different to represent" by "Everybody's trait value is allowed to vary in order to represent".

We have reworded this sentence to make it clearer. We didn't quite use the suggested wording as we felt that it could be misinterpreted as saying that trait values vary through time. Instead we have simply said "Different individuals can have different trait values, representing ..." (line 164)

+ page 6, line 179: It sounds a little awkward to say "Each individual in the dataset was assigned a gender manually based on their name". Perhaps this should be "Gender was inferred for each individual in the dataset from their name".

Done.

+ page 15, Figure 1: It took me a moment to realise that the data presented in the top row of figures were grouped by discipline in the same way as the bottom row. To increase clarity, it would be good practice to also display the discipline labels for the top row. The same comment applies to Figure 2 on page 16.

Additional labels have been added to the top rows of Figures 1 and 2.

+ page 17, Figure 3: Could the abbreviations "C-Ed", "Ed", "Pres", "LCA", "ECA", "Stud" be explained in the caption? This would be helpful for readers from non-English speaking backgrounds.

Done.

+ pages 19-22: Figures 1-4 are repeated without figure labels in the captions. Was this intentional?

We think this is a feature of the online journal submission system. It is not our intention to repeat the Figures.

Reviewer: 3

Comments to the Author(s)

I was undecided whether to list this review as Minor or Major revisions. I agree that it's useful to analyse the difference between male and female roles in scientific societies, and the authors have clearly gathered a lot of evidence, and completed the data with some modelling.

Overall, the manuscript was well-written, but precision was an issue in places, as discussed below. Presentation of the data, and conclusions drawn from the data presented, also need improvement.

I had the following concerns, which I believe need to be addressed before publication in this journal:

Point 1: Meaning of the data: I have some issues with the way the data was presented, and some of the conclusions drawn from the data. E.g.

1. Figure 1: This data is not normalized against the number of women in the societies, and hence doesn't show much useful information, I suggest remove. The data in Figures 2-3 is much more useful, since normalized against the numbers of women, but there are still serious issues, as outlined below. **We think it is relevant and important to show the raw data for the proportion of women in these categories, as well as the values normalised against the number of women in the societies. We have now explicitly clarified that this is what Figure 1 is showing (lines 214-215).**

2. Why no error bars or other symbols to demonstrate variability on Figure 2? Also bar graph seems a strange format for this kind of data?

After some consideration, we decided that adding error bars to Figure 2 would obscure the main messages of the graph, especially given the log scale for the vertical axes. Note that we do indicate which cases are statistically significant with red stars, and this corresponds to cases where the 95% confidence interval would exclude $r=1$. We have also rewritten the section describing the results in Figure 2 (see response to comment 4 below) so that these are now explained more clearly. We prefer to retain the bar graph format for compatibility with Figure 1, and for visual ease of grouping the four data points for each subject.

3. Figure 2 presents results by field and by country, Figure 3 pools them completely. What about when pooled just by field across all countries? Or by country across all fields? If pooling in this way doesn't show much, then even that is useful, it could be reported in the text. For example, (Ceci, Ginther et al. 2014) found that women actually do better in male-dominated fields of science, how does your data compare to their findings? This would be a useful comparison, because you have both male-dominated and non-male-dominated disciplines of science

We did test this but it did not lead to any interesting results. We have now reported this in the text on line 237-238.

4. One of your key conclusions is that women are under-represented in high status roles, I'm not convinced of this based on your data and discussion. You define chief editor as higher status than editor, but what about president? Furthermore, the differences data for chief and editors in figure 2 also don't easily lead me to conclude that men are more likely to be chief editor. See comment above also, about what this data would look like if pooled across discipline and /or nation. You also need some discussion of what is the status of editor vs president. Is president low status? Classify your positions each specifically as high or low status, consider even showing that classification on the plots **We agree this section did not describe our findings particularly well. We have now re-written this section to explicitly state the relative status of the various categories and how women's**

representation changes with this (lines 240-254). We also discuss separately the role of president (which is a special case carrying a mixture of status and administrative burden) more closely, and note the limitation of small sample size with respect to this role (Lines 255-260).

5. Figure 3: Your title includes the statement "...though this is improving", but that statement is not supported by the data, because you haven't presented data over time, or referred to other time/data

Thank you for pointing out this mistake – we have removed this statement from the caption.

6. You could easier make the x axis on this plot easier to read, using full names
As some of the labels are quite long (e.g. early career award), we have instead provided a key to the abbreviations with the full names in the Figure caption (as suggested by Reviewer #1).

Point 2: Imprecision

1. Introduction was in general well-written, but the first paragraph rather sensational, and imprecise in places, eg. "Science is sexist". I've read lots of papers about gender in science, but I can't agree with this statement, particularly because you haven't defined what you mean by sexist. If you want to make this statement, back it up with reasoning, including a definition of sexist

We have rewritten the first paragraph of the introduction, including an explanation in the opening sentence as to what we mean by "science is sexist", along with additional reasoning, evidence and references to back this statement up.

2. Throughout the introduction, you switch numerous times back and forth between the literature, and this study. To see this, do a search on "our analysis", and you will see that in some paragraphs you switch back and forth a number of times between what is known, and what you are trying to do here, which makes it hard for the reader to get a really coherent picture of the purpose of this paper, and its contribution

We agree the introduction switched confusingly between background literature and the current study. We have reordered the Introduction so that the background literature review comes first, followed by a description of what we set out to do in this study (final paragraph of Introduction). Note that in response to a comment by Reviewer #2, we have included a single sentence in the opening paragraph stating the aims of the study.

Other issues of precision in writing:

L65: Is what you call "Associate Editor" here what you call "Editor" in the figures? Best to keep terminology clear and consistent.

For clarity, we now use either Associate Editor or Chief Editor throughout the Methods and Results sections.

L95-97: reverse order of the sentence to make it clearer: women are getting more awards, but more likely to get those awards for teaching and service.

Done.

L149: reference needed

Reference added (Holmes et al, 2011).

L190: I presume you normalize by the number of women in the society, pooled by societies in that discipline across the country? Note that numbers in the society may not equal numbers in the discipline.

We normalise by a best estimate of the number of women in eligible positions. This is sourced from the benchmarking data for number of women in a given discipline-country grouping, rather than number of women in the society which is unknown. The definition of “eligible position” depends on the award or role. We have now clarified both these points with some additional text (lines 193-199). We have also noted that numbers in the society may not equal numbers in the discipline, but that this data though imperfect is the best available (lines 199-204).

L221-222: reference needed

References added.

L398: Second recommendation is written as “either or”, but these are 2 quite different statements with different consequences Ceci, S. J., D. K. Ginther, S. Kahn and W. M. Williams (2014). "Women in academic science: a changing landscape." *Psychological Science in the Public Interest* 15(3): 75-141.
1.

We have reworded this recommendation to make clear that soliciting a diverse range of nominees is the important thing.